# Paralinguistics-Aware Speech-Empowered Large Language Models for Natural Conversation

**Heeseung Kim**[1]    **Soonshin Seo**[2]    **Kyeongseok Jeong**[2]    **Ohsung Kwon**[2]    **Soyoon Kim**[2]
**Jungwhan Kim**[2]    **Jaehong Lee**[2]    **Eunwoo Song**[2,4]    **Myungwoo Oh**[2]    **Jung-Woo Ha**[2,3]
**Sungroh Yoon**[1,4,5*]    **Kang Min Yoo**[2,3,4*]

[1]Data Science and AI Lab, Department of ECE, Seoul National University
[2]NAVER Cloud    [3]NAVER AI Lab
[4]Artificial Intelligence Institute, Seoul National University
[5]ASRI, INMC, ISRC, and Interdisciplinary Program in AI, Seoul National University

## Abstract

Recent work shows promising results in expanding the capabilities of large language models (LLM) to directly understand and synthesize speech. However, an LLM-based strategy for modeling spoken dialogs remains elusive, calling for further investigation. This paper introduces an extensive speech-text LLM framework, the Unified Spoken Dialog Model (USDM), designed to generate coherent spoken responses with naturally occurring prosodic features relevant to the given input speech without relying on explicit automatic speech recognition (ASR) or text-to-speech (TTS) systems. We have verified the inclusion of prosody in speech tokens that predominantly contain semantic information and have used this foundation to construct a prosody-infused speech-text model. Additionally, we propose a generalized speech-text pretraining scheme that enhances the capture of cross-modal semantics. To construct USDM, we fine-tune our speech-text model on spoken dialog data using a multi-step spoken dialog template that stimulates the chain-of-reasoning capabilities exhibited by the underlying LLM. Automatic and human evaluations on the DailyTalk dataset demonstrate that our approach effectively generates natural-sounding spoken responses, surpassing previous and cascaded baselines. Our code and checkpoints are available at https://github.com/naver-ai/usdm.

## 1   Introduction

Large language models (LLMs) have gained significant traction thanks to emergent capabilities [1–5], such as few-shot in-context learning, complex reasoning [6, 7], and instruction-following [8]. These remarkable discoveries led to chat-enabled LLMs and generative personal assistants [9]. However, text-based agents are limited in usability due to their medium of interaction. Ideally, speech-enabled LLMs would recognize the user's emotional state or subtle nuance and generate spoken responses with prosody most appropriate to the user's context. Although automatic speech recognition (ASR) and text-to-speech (TTS) systems can be easily employed, the linguistic discrepancy between speech and text causes dialog inefficiencies and result in sub-optimal user experience [10, 11]. As such, systematically integrating the speech modality into LLMs can unlock speech interactivity while retaining LLMs' powerful capabilities.

---

*Corresponding authors: Kang Min Yoo <kangmin.yoo@navercorp.com>, Sungroh Yoon <sryoon@snu.ac.kr>

38th Conference on Neural Information Processing Systems (NeurIPS 2024).

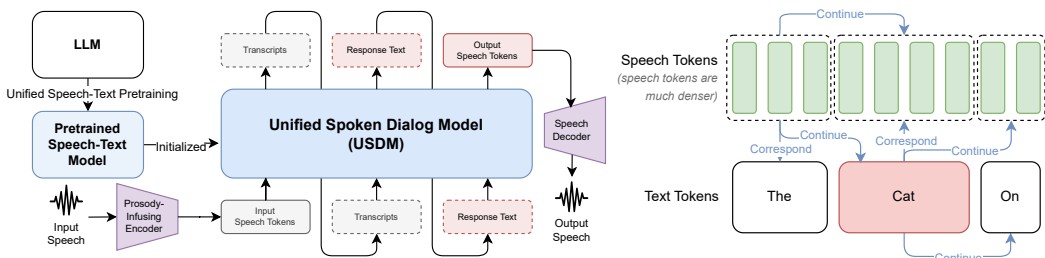

Figure 1: Overview of our spoken dialog modeling approach (Left). All possible self-supervised learning objectives from our speech-text pretraining scheme. (Right)

Recent advances have also spurred the idea of large foundational models (LFM) for other modalities (e.g., vision, speech, etc.) [12], unifying LLMs with the sensory spaces. In the vision domain, numerous works have explored interfacing pretrained language models with the visual modality [13–16]. More recently, Yu et al. [16] proposed a vision language model (VLM) based on a pretrained LLM that directly generates discrete vision tokens, which can be decoded into high-fidelity images. Such work shows that autoregressive language models can model the vision modality.

In the speech domain, while earlier work focused on text-less speech modeling [17–19], recent work has either taken inspiration from the LLM architecture to achieve speech synthesis [20] or incorporate pretrained language models into speech-understanding tasks [18, 21–23], where the model is limited to text outputs. More recent work explores the possibility of empowering pretrained LLMs to autoregressively generate discrete speech tokens for speech translation [24] and speech-instruction following tasks [25–27]. Despite these successes, more work is needed to understand whether LLMs are capable of generating speech, understanding, and incorporating paralinguistics that are appropriate and natural for the social context, especially in spoken dialog settings.

We introduce the Unified Spoken Dialog Model (USDM), a novel LLM-based approach for modeling spoken dialogs in an end-to-end fashion. We propose a novel speech-text pretraining scheme that promotes learning cross-modal distributional semantics, which is vital for imbuing LLMs with the ability to generate coherent speeches in spoken dialog modeling. In particular, based on the observation that any subsample of either speech or text in corresponding speech-text pairs form two types of relationships with the other modality (right side of Figure 1), we formulate a large number of combinations of training objectives that theoretically benefits all speech-text tasks, including spoken dialog modeling. We then fine-tune our pretrained speech-text model with spoken dialog data, breaking the speech-to-speech modeling problem into intermediary steps that are easier for the underlying pretrained LLM to handle (Figure 1). To enhance the effects of our speech-text pretraining and spoken dialog modeling, in addition, we adopt the prosody-infused speech tokenization scheme based on the discovery that the speech token, previously used to convey semantic information, also contains prosody information.

We demonstrate that USDM outperforms baselines in spoken dialog modeling for the DailyTalk dataset. We further validate the effectiveness of our pretraining and fine-tuning schemes through comprehensive ablation studies. Along with various analyses, we highlight the capabilities of USDM with diverse samples on our demo page.[2] Our contributions are as follows.

- We propose a unified pretraining strategy for modeling the comprehensive relationship between the speech and text modalities that is especially effective for downstream speech-to-speech spoken dialog generation.

- We present an extensive spoken dialog modeling framework detailing the discrete speech tokenization scheme utilizing a pair of a prosody-infusing encoder and a decoder. Additionally, we propose an LLM-based modeling strategy for generating natural-sounding and semantically coherent dialog responses in an end-to-end fashion.

- Our work establishes the foundation for speech-enabled chat-based LLMs, showcasing a prototype that not only maintains the LLM's ability to generate dialog responses but also enhances LLM with speech-interaction capabilities.

---

[2]Demo: https://unifiedsdm.github.io/

## 2 Related Work

**Discrete Speech Representations.**   To construct spoken language models (SLM), various discrete speech representations have been utilized in previous works [17, 20, 28–30]. These representations are primarily categorized into two types: tokens based on speech self-supervised representations [17, 30] and neural audio codecs [20].

A discrete token based on speech self-supervised representation [17, 31] is obtained by $k$-means clustering of the intermediate representation from a speech self-supervised model. These tokens, often called acoustic units, are typically encoded with a frequency range of 25Hz to 50Hz. The amount of speech information within the compressed discrete speech token is determined by the number of clusters, denoted as $k$. With a relatively small value for $k$, many works have preserved the semantic information in the tokens and utilized these to construct SLMs [32, 33].

Neural audio codecs, another type of discrete token, capture both semantic and paralinguistic information of speech [34–38]. A speech encoder and decoder are trained using an autoencoder architecture with residual vector quantizer for the encoder output [39]. This representation includes most of the perceptual information of audio and is widely used for audio synthesis [20, 40–42].

**Spoken Language and Dialog Models.**   Many studies have recently explored spoken language modeling to address a variety of tasks involving speech and text [23, 43–45]. Various works tackle tasks such as automatic speech recognition [27, 46–49], spoken question answering [50–52], and speech-to-text translation [27, 48, 53], which process speech as input and output text. Conversely, there are also emerging works focused on tasks like speech synthesis [20, 26, 28, 54, 55], where text is used as input to generate speech output. Early SLMs that process speech as input and output are trained solely based on speech data without language models [17, 19]. With the advancement of LLMs, several studies aim to construct SLMs that extend language models to handle both speech input and output. These studies are primarily proposed for speech modality pretraining [30, 56, 57], or introduced in specific tasks such as speech-to-speech translation [24, 58–60] and spoken conversation modeling [25, 61].

Recently, several works have been proposed for spoken dialog modeling with speech input and output [19, 25, 62]. Nguyen et al. [19] develop a decoder-only transformer model trained from scratch, designed for modeling conversations between two speakers. In contrast, Lin et al. [62] adopt a cascaded approach for spoken dialog modeling that consists of separate ASR, an LLM-based emotion-aware text dialog model, and emotional TTS components. Zhang et al. [25] build SLMs on top of a pretrained LLM with objective functions designed for ASR and TTS tasks.

Among the previous works, end-to-end pipelines [19, 25] that focus solely on speech-only training or leverage simple cross-modal objectives for speech-text pretraining fail to fully utilize the capabilities of pretrained language models. Additionally, cascaded models [62], which use separate ASR and TTS for spoken dialog, need explicit labels to incorporate paralinguistic features. This label dependency makes data collection challenging and limits the models to representing label-definable non-verbal cues. Furthermore, the error propagation inherent in the cascaded pipeline [63] increases their susceptibility to compounded errors.

## 3 Our Approach

In this section, we describe the components that enable coherent and prosodic spoken dialog modeling, distinguishing our research from previous works. We first explain the discrete speech representation used for spoken dialog modeling in Section 3.1, demonstrating its suitability for prosody modeling. We then propose a unified speech-text pretraining scheme that extends the capabilities of the pretrained LLM into the domain of spoken language modeling in Section 3.2. Finally, in Section 3.3 and 3.4, we introduce USDM, a spoken dialog model fine-tuned with a multi-step spoken dialog template, and the speech decoder that restores the output speech token to a raw waveform.

### 3.1 Speech-to-Unit Encoder

To model natural speech conversations, the speech representation must contain not only the content of the speech but also paralinguistic features such as emotions, which are crucial for conversation. We

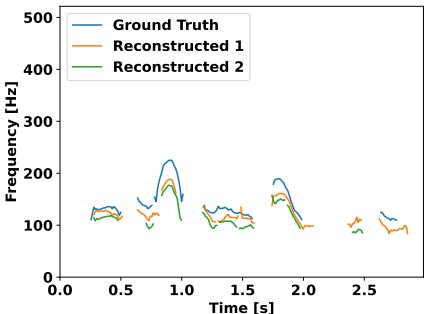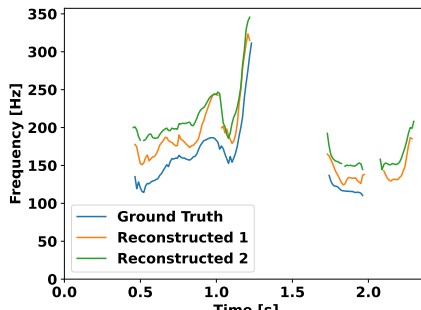

Figure 2: Pitch contour of the original audio and the audio reconstructed from extracted acoustic units. Due to the stochastic nature of the reconstruction model, we attempt reconstruction twice, demonstrating that the pitch variation closely mirrors the ground truth.

adopt acoustic units as speech tokens that are derived from $k$-means clustering of a self-supervised model's intermediate speech representation, which is known to predominantly capture content and pronunciation [31, 59]. The information captured by an acoustic unit token varies depending on the number of clusters; the greater the number of clusters, the more encoded information. Hence, among publicly available schemes, we consider the acoustic unit tokenization scheme with the largest vocabulary size, $k = 10,000$ [59]. We then analyze whether the tokens derived from this scheme contain non-verbal information.

The acoustic unit extractor used in SeamlessM4T [59], first resamples the speech to a sampling rate of 16kHz and then feeds it to the XLS-R [64], obtaining 50Hz intermediate continuous representations. Unit sequences are extracted by clustering these representations into 10,000 clusters, determining the vocabulary size of the speech tokens. Using this unit extractor, we conduct two experiments to investigate features captured in the unit sequence besides the semantic content. First, we perform unit emotion recognition tasks with speech emotion recognition data to ascertain whether the unit sequences contain paralinguistic information. Next, we train a separate unit-to-speech reconstruction model and use this model to compare the original and reconstructed speech, investigating the information encoded in the unit sequences.

For the unit emotion recognition task, we train 3-layer transformer-based emotion classifiers using acoustic units on CREMA-D [65], which is a speech emotion recognition dataset with six emotion categories. If the units lack paralinguistic information, the classification accuracy would approximate the probability of random guessing, which is $16.6\%$. However, we observe that the classification accuracy is $60.8\%$, indicating that the acoustic units contain emotional cues.

To further investigate the paralinguistic information contained in the units, we train a separate unit-to-speech reconstruction model using 54,000 hours of speech data. The reconstruction model is trained using the architecture of Voicebox, one of the zero-shot stochastic TTS models [66]. Unlike Voicebox, which takes text and reference speech as inputs for adaptation, our model is trained to generate speech solely from a unit sequence without any reference speech. The comparison between the original audio and the speech reconstructed from the extracted units shows that while the timbre and absolute pitch of the reconstructed speech differ, the pitch variation has a similar trend, as shown in Figure 2. This implies a crucial role in conveying non-verbal characteristics such as emotions, which closely match the original audio. Additionally, we have uploaded samples of several ground truth audios and corresponding reconstructed audios on our demo page.

Through these two experiments, we confirm that the acoustic units, commonly known to primarily encode semantic information, also contain a significant amount of paralinguistic information, such as emotions and pitch variations. We adopt this speech tokenization scheme for our speech-text pretraining and spoken dialog fine-tuning to help capture non-verbal cues in spoken conversations. More detailed descriptions of these experiments are provided in Appendix A.3.1 and A.3.3.

### 3.2 Unified Speech-Text Pretraining

In this section, we introduce a unified speech-text pretraining scheme that extends pretrained LLM to speech-text cross-modality. Our overall speech-text pretraining scheme is in Figure 3.

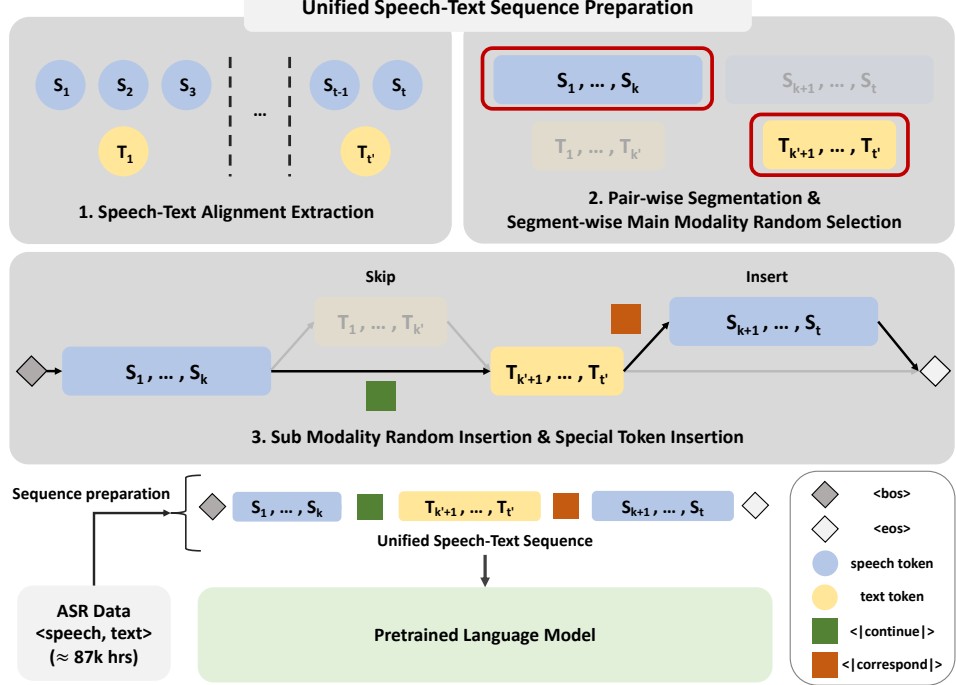

Figure 3: The overall speech-text pretraining scheme.

For pretraining the speech-text model, we utilize Mistral-7B [67] as a pretrained LLM. To its existing vocabulary, we add 10,000 unit tokens and 2 special tokens, which will be described later, reinitializing only the embedding weights of these new tokens. We pretrain the speech-text model with approximately 87,000 hours of English ASR data. Each <speech, text> pair is used to create an interleaved speech-text sample $\mathcal{I}_j = i_{1,j}, i_{2,j}, ..., i_{||\mathcal{I}_j||,j}$ to model various cross-modal relationships. Here, $i_{k,j}$ can be either an acoustic unit token, a text token, or a special token, and we construct the sequence $\mathcal{I}_j$ with a proposed per-sample speech-text interleaving method in the following paragraph. Given the dataset $\mathcal{D} = \{\mathcal{I}_1, \mathcal{I}_2, ..., \mathcal{I}_{||\mathcal{D}||}\}$, the objective of our pretraining scheme is as follows:

$$\mathcal{L}(\theta) = -\Sigma_{j=1}^{||\mathcal{D}||}\Sigma_{k=1}^{||\mathcal{I}_j||}\log p(i_{k,j}|i_{<k,j};\theta), \qquad (1)$$

where $\theta$ refers to the parameters of LLM and the embedding weights of the newly added tokens.

When constructing a spoken language model by extending pretrained LLM, relying on a specific task, such as ASR [25], TTS [25], uni-modal [30] and cross-modal continuation task [44, 56], despite its large amount of dataset, may limit the model's capabilities to only those predefined relationships. To build a comprehensive speech-text model capable of both receiving and generating speech, we reinterpret the cross-modal relationship in terms of continuation or correspondence as shown in Figure 1. Our proposed method, which focuses on this redefined relationship, is capable of generating diverse speech-text interleaved sequences, ensuring the model can handle complex speech-text interactions.

**Speech-Text Alignment Extraction** We first extract word-level alignments of the speech and its transcript using the Montreal Forced Aligner [68]. These alignments yield speech time intervals for each word, which we then convert into index intervals of unit sequences at a resolution of 50Hz.

**Pair-wise Segmentation and Segment-wise Main Modality Random Selection** Using the intervals, we divide each unit and text pair into $N$ segments. Subsequently, from each of these segments, we randomly sample data of only one modality, either unit or text. A large value of $N$ may lead to each segment containing short acoustic units and text sequences, which poses challenges in modeling unimodal text and unimodal unit sequences. To mitigate this issue, we dynamically set the value of $N$ based on the speech duration, $N = \lfloor S/10 \rfloor + 1$, where $S$ is the speech length measured in seconds.

**Sub Modality Random Insertion and Special Token Insertion** This segmentation and selection process allows us to generate a unified cross-modal interleaved sequence with continuation relationships.

For correspondence relationship modeling, data from the non-selected modality in each segment is inserted with a 50% probability after the pre-selected modality data. Additionally, to indicate the relationship between speech and text tokens, we introduce two special tokens: `<|correspond|>` and `<|continue|>`. The former indicates that a token of the corresponding remaining modality will follow, while the latter indicates that a token of the subsequent position will follow. These tokens are added to the sequence only where the modality of the data changes.

Through this procedure, we can obtain interleaved speech-text sequences $\{\mathcal{I}_j\}_{j=1,\ldots,||\mathcal{D}||}$. These sequences enable our speech-text model to perform not only unimodal modeling but also comprehensive cross-modal modeling. These interleaved sequences are utilized in Eq. 1 for the pretraining.

### 3.3 Unified Spoken Dialog Model

We construct the Unified Spoken Dialog Model (USDM) by fine-tuning our speech-text model with spoken dialog data, with an overview presented as shown in Figure 1. The basic template for spoken dialog fine-tuning involves directly modeling the response speech tokens from the input speech tokens. However, we adopt a template designed to fully exploit the capabilities of the pretrained LLM.

Inspired by the step-by-step reasoning mechanism employed by LLMs [69], existing works [24, 25, 57] use text to bridge the speech. Similarly, instead of directly modeling output speech from input speech, our model transcribes the speech, generates the response text, and produces the corresponding speech in an end-to-end pipeline. The insertion of the text-related tasks between speech inputs and outputs allows the model to benefit from the pretraining LLM and chained reasoning over the intermediary modality [6]. Since each stage in the pipeline attends to all input and output tokens generated in prior stages, our approach is more robust against transcription errors and better at generating contextually relevant spoken responses than if it were carried out in independent modules (i.e., the cascaded approach), which we will discuss further in Section 4.3.

The supervised fine-tuning template we use is shown in Figure 5 in the Appendix. We calculate the loss using Eq. 1 only for the input transcript, answer text, and answer unit part, as highlighted in Figure 5.

### 3.4 Unit-to-Speech Decoder

We train the unit-to-speech model using the Voicebox architecture [66] to reconstruct speech from units. Voicebox is a zero-shot TTS model that takes text and reference speech as inputs to perform personalized TTS. Unlike the reconstruction model used in Section 3.1, we leverage not only unit sequences but also reference speech to adapt and perform zero-shot unit-to-speech reconstruction. Our model utilizes the reference speech and the paralinguistic features contained in the units to generate prosodic spoken responses of the target speaker. For clarity, we refer to this unit-to-speech model as unit-Voicebox. More details of our decoder are in Appendix A.3.3.

## 4 Experiments and Results

### 4.1 Model Comparisons

#### 4.1.1 Training Details and Baselines

We compare USDM to 3 baselines, From Scratch, Cascaded, and SpeechGPT [25], on DailyTalk [70]. DailyTalk comprises 20 hours of spoken dialog data with a sampling rate of 22,050 Hz, involving one male and one female, and we describe further details in Appendix A.4.1. We also present the models used for each component of USDM and the baselines in Table 4 in the Appendix.

**USDM.** For speech-to-unit module, we adopt the official checkpoint of XLS-R [64] and a quantizer with $k = 10,000$ [33], trained on 436K hours of multilingual speech data. As a speech decoder, we follow the architecture and hyperparameters of Le et al. [66] and train the unit-Voicebox on the English subset of Multilingual LibriSpeech [71] and GigaSpeech [72] for 10 epochs using 64 NVIDIA A100-40GB GPUs, with a batch size of 256. We use the Adam optimizer [73] with a learning rate of $10^{-4}$. We utilize the official checkpoint of BigVGAN [74] as our vocoder.

Our proposed unified speech-text pretraining is conducted using 512 NVIDIA A100-40GB GPUs, with a global batch size of 1,024 for 8,000 iterations. For pretraining, we utilize approximately 87,000 hours of English ASR data; the English subset of Multilingual LibriSpeech [71], People's Speech [75], GigaSpeech [72], Common Voice 15.0 [76], and the English subset of Voxpopuli [77]. The data used for pretraining is packed to a maximum sequence length of 8,192. For spoken dialog modeling, we fine-tune a speech-text model with a global batch size of 64 for 5 epochs. We use linear learning rate scheduling with a peak learning rate of $2 \cdot 10^{-5}$ for both pretraining and fine-tuning.

**From Scratch.** The From Scratch model is nearly identical to the USDM but excludes speech-text pretraining. Specifically, we fine-tune the pretrained Mistral-7B model directly on spoken dialog data, with the hyperparameters identical to those of the USDM.

**Cascaded.** We include a Cascaded model, which employs separate ASR and TTS models, as a baseline for comparison. For the ASR model, we use the official checkpoint of *whisper-large-v3* [78], which is trained on 5M hours of speech data. For the speech synthesis model, we train Voicebox with text input using the same hyperparameters and datasets as unit-Voicebox. As the LLM, we utilize the transcript of the spoken dialog dataset to create text dialog data and fine-tune the Mistral-7B on this data using the same hyperparameters as the USDM.

**SpeechGPT.** For SpeechGPT [25], we use the official implementations and checkpoints for the speech-to-unit module, spoken language model, and speech decoder module. Specifically, we fine-tune *SpeechGPT-7B-cm*, a pretrained speech-text model, with DailyTalk for a fair comparison.

### 4.1.2 Evaluation and Comparison Results

We conduct various evaluations on the spoken responses generated for the given spoken dialogs. When generating samples for evaluation, we adopt a sampling scheme with top_k = 40, top_p = 0.7, and temperature = 0.3, except for SpeechGPT, where we use their own strategy. For Voicebox and unit-Voicebox, we utilize the speech from the previous turn as the reference speech. While SpeechGPT generates audio at 16kHz, other models synthesize speech at 22,050Hz. For a fair comparison, all audio samples are resampled to 16kHz and volume normalized to -27dB for evaluation.

To compare the overall preference of our model and the baselines, we conduct a human preference test via Amazon Mechanical Turk. Given a randomly selected 50 spoken dialogs from the test split of a dataset, we instruct the evaluators to compare the spoken response of our model and the baseline, considering the comprehensive aspects such as naturalness, prosody, and semantic coherence. To evaluate the prosody and the naturalness, we additionally measure the 5-scale prosody mean opinion score (P-MOS) and the 5-scale mean opinion score (MOS) through Amazon Mechanical Turk. As explained in Section 3.3, our model first generates the text to be spoken before generating a spoken response, which allows us to fix the content of the generated speech by predetermining the text. Unlike the aforementioned human preference test, to focus solely on the prosody and naturalness, respectively, we provide the ground truth response text to the model to ensure consistency in the content of the output speech, thereby preventing difficulties in evaluations that may arise from variations in content. Instructions and detailed descriptions of our evaluations are in Appendix A.4.2.

Furthermore, to evaluate the content appropriateness of spoken responses, we first generate the spoken responses of all models given the spoken dialogs for the entire test set. We then pass these samples through the ASR model, *whisper-large-v3*, to calculate METEOR and ROUGE-L scores, which are widely used in various NLP tasks such as text summarization and are commonly employed to measure performance in dialog modeling [79, 80]. We also conduct a GPT-4-based [1] preference test [81] between the transcribed texts of our model and all baselines.

We also measure the Word Error Rate (WER) for the speech-to-text part and text-to-speech part of each model. For the speech-to-text part (STT WER), we calculate the WER across the entire test set. We use the outputs from the *whisper-large-v3* model for the Cascaded baseline, while the remaining end-to-end pipelines are assessed using the intermediate transcribed text of each model. For the text-to-speech part (TTS WER), similar to our MOS and P-MOS evaluations, we calculate the WER using generated samples given the randomly selected 50 spoken dialogs and the corresponding ground truth written-form response. We generate each spoken response 5 times and report the average WER. For measuring TTS WER, we utilize the *whisper-large-v3* model as the ASR model.

The results are presented in Table 1 and 2. In human preference tests that consider comprehensive factors, our model is preferred similarly to the Ground Truth and demonstrates superior preferences

Table 1: Human evaluation results of our model and the baselines. We report the MOS and P-MOS scores with a 95% confidence interval.

| Method | Overall | | | Acoustic | |
|---|---|---|---|---|---|
| | *win* | *tie* | *lose* | MOS | P-MOS |
| Ground Truth | 45.9% | 8.0% | 46.1% | $4.51 \pm 0.05$ | $4.35 \pm 0.05$ |
| USDM | – | – | – | $4.31 \pm 0.07$ | $4.31 \pm 0.06$ |
| Cascaded | 55.3% | 4.9% | 39.8% | $4.26 \pm 0.07$ | $4.22 \pm 0.07$ |
| From Scratch | 53.3% | 7.6% | 39.1% | $3.71 \pm 0.11$ | $3.65 \pm 0.10$ |
| SpeechGPT [25] | 53.8% | 6.9% | 39.3% | $4.08 \pm 0.09$ | $4.04 \pm 0.08$ |

Table 2: GPT-4 evaluation and quantitative results of our model and the baselines.

| Method | Semantic | | | | | WER | |
|---|---|---|---|---|---|---|---|
| | *win* | *tie* | *lose* | METEOR | ROUGE-L | STT | TTS |
| Ground Truth | 32.7% | 19.6% | 47.7% | – | – | – | 2.2% |
| USDM | – | – | – | 13.1 | 15.7 | 7.4% | 2.0% |
| Cascaded | 42.7% | 24.6% | 32.7% | 12.5 | 15.0 | 3.8% | 1.3% |
| From Scratch | 79.7% | 10.1% | 10.2% | 8.6 | 10.6 | 58.1% | 64.0% |
| SpeechGPT [25] | 61.0% | 13.1% | 25.9% | 9.9 | 11.8 | 12.4% | 23.2% |

compared to the baselines ($p$-value $< 0.05$ from the Wilcoxon signed-rank test). For the semantic aspect, our USDM outperforms the baselines in both quantitative evaluations and the GPT-4-based preference test ($p$-value $< 0.05$). We also observe that our model surpasses the baselines in the P-MOS evaluations ($p$-value $< 0.05$), which measure the prosody naturalness of the speech given spoken dialog. Notably, the USDM shows superior prosody compared to the Cascaded model. These results demonstrate that our model effectively incorporates prosody information in the spoken language model and is capable of generating spoken responses with content well-aligned to input speech.

We also confirm that cross-modal pretraining is essential to leverage the capabilities of LLM. We observe that the From Scratch model, which directly models spoken dialog without pretraining, tends to overlook the bridging text and generates a spoken response that does not correspond to the pre-generated written response, thus negatively impacting its performance. This results in worse TTS and STT WERs and adversely affects the P-MOS and MOS, which are based on the prosody and naturalness of the spoken response given the transcript. This result indicates the difficulty of transferring the capabilities of text models to spoken dialog modeling without cross-modal pretraining.

## 4.2 Ablation Studies

### 4.2.1 Pretraining Schemes

In this section, we compare the effects of correspondence and continuation modeling, which are crucial to our pretraining method. We consider three additional pretraining schemes. **Setup 1** uses an interleaved unit-text sequence without a correspondence relationship, relying solely on continuation, similar to previous works [44, 56]. **Setup 2** utilizes data that maintains only a correspondence relationship, as seen in Zhang et al. [25]. **Setup 3** is similar to our fine-tuning approach, interleaving speech with its transcript and subsequent text and speech, as proposed in Nachmani et al. [57]. All setups are trained in the same way as our speech-text pretraining, with details in Section A.5.2.

We evaluate these pretrained models on sequence modeling and spoken dialog modeling tasks. Performance is first assessed by measuring perplexity (PPL) of various speech-text sequences from the `test-clean` subset of the LibriSpeech dataset [82]. We construct six types of interleaved sequences: unimodal sequences for both unit (1) and text (2), sequences with unit followed by their corresponding text (3) and vice versa (4) to evaluate correspondence relationships, and sequences generated by dividing the unit and text in half, combining the first half's unit with the remaining half's text (5), and the first half's text with the remaining unit (6) for assessing continuation. We then calculate the average PPL for all combinations by taking the logarithm of each subsequent modality's PPL within each sequence type, averaging these logarithmic values, and then applying

Table 3: Results of the ablation studies on the pretraining and fine-tuning schemes. For PPL, we report the average PPL for each modality across the six combinations described in the text.

| Method | Pretraining | | Spoken Dialog Modeling | | | |
|---|---|---|---|---|---|---|
| | Text PPL | Unit PPL | STT WER | TTS WER | METEOR | ROUGE-L |
| Ours | 6.886 | 4.813 | 7.4% | 2.0% | 13.1 | 15.7 |
| Setup 1 | 14.485 | 5.261 | 57.8% | 82.1% | 8.9 | 10.6 |
| Setup 2 | 31.679 | 5.619 | 11.2% | 2.5% | 12.5 | 15.1 |
| Setup 3 | 21.392 | 5.146 | 7.3% | 2.0% | 12.7 | 15.4 |
| S1 → S2 | – | – | – | – | 6.5 | 7.7 |

the exponential function. For spoken dialog modeling, we fine-tune each model with DailyTalk and measure STT WER, TTS WER, METEOR, and ROUGE-L, as described in Section 4.1.2.

We present the average PPL of each modality in Table 3 and the PPL for each combination in Table 9 in the Appendix. Our model demonstrates superior average PPL across both modalities. Focusing solely on either correspondence or continuation relationships, or on specific templates tends to make the model specialize in certain objectives but hinders its ability to model diverse relationships effectively. Our proposed unified speech-text pretraining scheme performs uniformly well without being overly focused on specific relationships. We also show in Table 3 that our speech-text model is beneficial to spoken dialog modeling, as evidenced by the WER, METEOR, and ROUGE-L scores. Particularly, Setup 1, which lacks correspondence relationship pretraining, exhibits significantly higher STT and TTS WERs, resulting in deteriorated semantic performance in spoken responses.

### 4.2.2 Fine-tuning Schemes

As explained in Section 3.3, USDM first models the input and output text as a bridge when given a speech input before generating the spoken response. To demonstrate this approach, we train a spoken dialog model that models the speech output directly from the speech input (S1 → S2). We evaluate the generated response speech through METEOR and ROUGE-L scores with the same samples described in Section 4.1.2, and the results are shown in Table 3. We find that intermediate text modeling in spoken dialog generation helps generate appropriate spoken responses. This confirms that the process of generating text before speech leverages the capabilities of the pretrained model effectively.

### 4.3 Analysis on Input Modality

As seen in Table 2, the Cascaded model exhibits a lower ASR WER compared to USDM. This is due to the separate ASR model used in the Cascaded model, *whisper-large-v3*, which has been trained on approximately 5 million hours of speech data. However, in terms of the semantics of the spoken response, our model outperforms the Cascaded model.

Similar to previous works that show the advantages of end-to-end pipelines over cascaded approaches in several tasks [83–86], USDM leverages input speech to generate more semantically coherent answers. To empirically verify that our generated text responses utilize information from both the preceding transcript and the input speech, we plot the attention maps for each layer, as illustrated in Figure 4. We calculate the probability that each token in the generated response attends to each token in the input unit sequence and corresponding transcript by averaging the probabilities across all heads of the attention modules for each layer. Subsequently, we aggregate these probabilities for input unit tokens to compute a cumulative probability for the speech input, and similarly for text tokens relative to the transcript. As shown in Figure 4, our generated tokens attend not only to the transcribed text but also to the speech input, indicating that our model benefits from the speech input.

In Figure 4, we also observe that the generated responses notably attend to the transcribed text. To analyze the impact of more accurate transcription on model performance, we substitute the model-generated input transcript with the ground truth transcript in the middle of the inference of USDM and measure the METEOR and ROUGE-L scores for the generated spoken responses. The measured scores are 13.6 and 16.2, respectively, surpassing our model's previous results of 13.1 and 15.7 in Table 2. This confirms that enhancing the accuracy of unit-to-text conversion in USDM also improves the semantic coherence of the spoken responses.

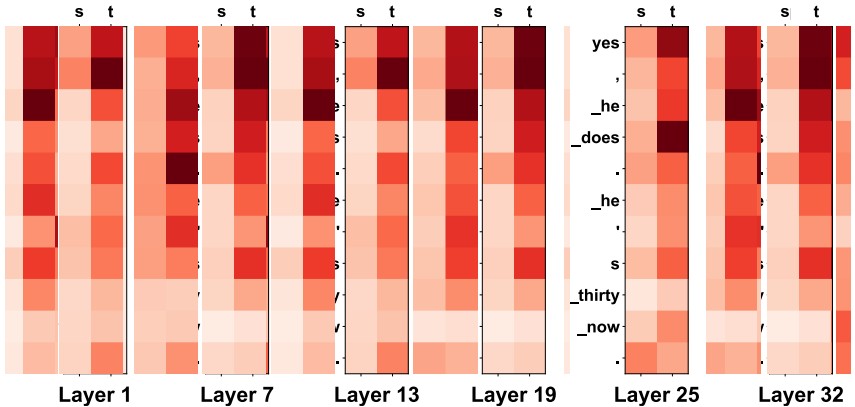

Figure 4: Attention maps between the generated responses of the USDM and the input speech (s) and its transcribed text (t). Input speech: "Oh, I can't believe it. He looks very young."

## 5    Conclusion

In this work, we presented USDM, a model synthesizing spoken dialog responses enriched with natural prosody. We proposed a novel speech-text pretraining scheme that models the comprehensive relationship between speech and text, which proves beneficial for spoken dialog modeling. Our approach is complemented by leveraging an acoustic unit tokenization scheme that preserves prosodic information, coupled with a supporting pair of an encoder and a decoder. We showed that USDM outperforms the baselines regarding content, prosody, and naturalness as a spoken response for the DailyTalk dataset. Additionally, we demonstrated that our pretraining and fine-tuning scheme benefits the USDM in modeling spoken dialog through ablation studies. Various samples for diverse scenarios in our demo page also showcased the capabilities of USDM. We believe that USDM has laid the groundwork for extending the conversational capabilities of LLMs to the voice domains.

Despite these advantages, our model has several limitations and areas for improvement. Firstly, the exploration of datasets and models used for pretraining is limited. Further investigation is necessary to determine which data are crucial for our pretraining scheme and to explore whether our pretraining scheme could be effective with other LLMs beyond Mistral-7B. Secondly, building a spoken dialog model capable of directly generating spoken responses from input spoken dialog without the need for cross-modal chaining can be a promising direction. Next, the current pretraining scheme is based on tens of thousands of hours of English data, and it has limitations when applied to other languages with relatively smaller amounts of speech data compared to English. We plan to expand our model to a variety of languages in addition to English. Lastly, we also plan to investigate whether our pretraining approach is beneficial for other speech-text tasks beyond spoken dialog modeling.

## Acknowledgments and Disclosure of Funding

This work was supported by SNU-Naver Hyperscale AI Center, Institute of Information & communications Technology Planning & Evaluation (IITP) grant funded by the Korea government(MSIT) [NO.RS-2021-II211343, Artificial Intelligence Graduate School Program (Seoul National University)], National Research Foundation of Korea (NRF) grant funded by the Korea government (MSIT) (No. 2022R1A3B1077720, No.2022R1A5A7083908), and the BK21 FOUR program of the Education and Research Program for Future ICT Pioneers, SNU in 2024.

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

# A  Appendix

## A.1  Audio Samples

We have included various audio samples from our experiments on our demo page.[3] Furthermore, to demonstrate the applicability and potential of our model, we have added several samples of USDM fine-tuned on the Expresso dataset [87], which contains emotionally rich spoken dialog data, and the Fisher dataset [88], a telephony conversation dataset between two speakers recorded at 8,000Hz, totaling approximately 1,960 hours from 11,971 speakers.

The Expresso dataset comprises 41 hours of emotionally expressive speech data from 4 speakers. Of this, 11 hours consist of simple reading styles, while 30 hours are improvised dialogs between two speakers. We use this dialog data to train our USDM, noting that these improvised dialogs lack corresponding transcripts. Therefore, we create transcripts using *whisper-large-v3* and utilize these to train the USDM. We observe that the trained model often failed in the ASR and TTS parts due to numerous inaccuracies in the transcripts generated by the ASR model, thus not using this data in the main experiments of our paper. Instead, we provide selected samples from the model trained on this data on our demo page.

In addition, we extend the spoken dialog templates used in training USDM to multi-turn scenarios to show the capability of USDM for modeling multi-turn dialogs. We explore this possibility by fine-tuning our speech-text model with Fisher [88]. We split the train and test sets with no overlapping speakers to show the possibility of unseen speakers' spoken dialog modeling. We have included samples of USDM's generated responses for multi-turn dialogs with unseen speakers of Fisher on our demo page. These samples demonstrate the potential of USDM for multi-turn spoken dialog modeling.

## A.2  Broader Impacts

This paper proposes a spoken dialog model designed to generate spoken responses to the speech inputs. Similar to the field of natural language processing, the dialog model might exhibit biases in its outputs, which stem from the training dataset. Such biases could unintentionally lead to the generation of synthesized voices that are biased. Furthermore, there are ethical considerations regarding the possible misuse of high-quality speech synthesis models, such as in voice phishing scams.

Despite these concerns, research into spoken dialog models can yield several positive effects. Unlike text, voice interactions are capable of conveying non-verbal information, allowing us to build conversational agents that consider users' emotions, which are challenging to capture with text-based dialog models. Furthermore, by introducing spoken language as an additional means of communication, we offer an alternative to text-based chatbots for individuals facing difficulties in reading and writing. Similar to text-based chatbots, which have rapidly evolved and brought convenience to daily life, we expect that, with careful consideration of ethical issues, research and development in spoken dialog models will significantly benefit everyday life and a wide range of industries.

## A.3  Additional Details for Our Approach

### A.3.1  Emotional Cues in Acoustic Units

In Section 3.1, we demonstrate through two experiments that the acoustic units extracted from XLS-R [64] contain paralinguistic features. The first experiment utilize CREMA-D [65], to perform a unit-to-emotion recognition task, to check whether the unit sequence contains information beyond content. Additionally, we train a unit-to-speech reconstruction module and use it to compare the original audio with the audio reconstructed from the extracted units, showing that the unit contains information about pitch variations, as shown in Figure 2 and several samples on our demo page. Further details about the trained unit-to-speech reconstruction module are provided in Section A.3.3.

For emotion recognition, we train a 3-layer transformer-based emotion classifier using cross-entropy loss. We utilize the CREMA-D dataset, consisting of 7,442 audios from 91 actors, categorized into six emotions: Anger, Disgust, Fear, Happy, Neutral, and Sad. We split the data into training, validation,

---

[3]Our demo is available at https://unifiedsdm.github.io/.

```
Below is a conversation between the user and the
agent. Each turn includes the user's speech and
its corresponding transcript, along with the
agent's response text and the corresponding
speech

### User
speech token <|correspond|> text token
### Agent
text token <|correspond|> speech token
```

```
Below is a conversation between the user and the
agent. Each turn includes the user's text, along
with the agent's response text.

### User
text token
### Agent
text token
```

Figure 5: Fine-tuning template for single-turn spoken dialog modeling. Left is the template used for training spoken dialog models (USDM, From Scratch), while the right is the template for training a text dialog model (Cascaded).

Table 4: Models for each component of the USDM and the baselines.

| Model | ASR Model | Speech Encoder | LLM | Speech Decoder | TTS Model |
|---|---|---|---|---|---|
| USDM | – | XLS-R | Mistral-7B | unit-Voicebox + BigVGAN | – |
| From Scratch | – | XLS-R | Mistral-7B | unit-Voicebox + BigVGAN | – |
| SpeechGPT | – | mHuBERT | Llama-7B | unit-HiFi-GAN | – |
| Cascaded | *whisper-large-v3* | – | Mistral-7B | – | Voicebox + BigVGAN |

and testing sets in a ratio of $70\%$, $15\%$, and $15\%$, respectively, ensuring an equal number of samples for each emotion in both the validation and test sets.

### A.3.2 Templates for Fine-tuning

Figure 5 illustrates the template of single-turn spoken dialog and text dialog we used for fine-tuning. We perform loss calculations only on the highlighted part. USDM refers to our model, and From Scratch and Cascaded are the baselines we used as described in Section 4.1.1.

### A.3.3 Voicebox

Voicebox [66] is a Flow-Matching-based zero-shot TTS model [89] that generates a mel-spectrogram from input text and reference speech. During training, Le et al. [66] utilize the Montreal Forced Aligner (MFA) [68] to extract the alignment between the phoneme sequence and the mel-spectrogram. This alignment is utilized to calculate the duration for each phoneme, allowing Le et al. [66] to expand the phoneme sequence to match the length of the mel-spectrogram. Voicebox is trained to produce a mel-spectrogram from an expanded phoneme sequence of equivalent length. Le et al. [66] additionally train a duration predictor to predict the duration of each phoneme in the sequence, which is used for inference.

We train three variants of the Voicebox for different purposes, all using the English subset of the Multilingual LibriSpeech [71] and GigaSpeech [72], totaling 54k hours of ASR data. We perform inference using a total of 50 timesteps with a classifier-free guidance scale of 1. We train the model to produce mel-spectrograms of approximately 86Hz, with the same configuration as the official 22,050Hz checkpoint of BigVGAN [74]. The output mel-spectrograms of all Voicebox models are converted into 22,050Hz speech through BigVGAN. This section will detail each of these 3 cases.

**Variant for Analyzing Non-Verbal Cues in Acoustic Unit** As described in Section 3.1, we train a unit-to-speech reconstruction model to investigate the non-verbal information contained within the unit. This model differs from Voicebox in that it utilizes unit sequence instead of text input and does not use reference speech during training and inference. We do not train a separate duration predictor; instead, we upsample the 50Hz unit sequence to 86Hz to match the length of the mel-spectrogram. Note that this variant is used only for analyzing non-verbal information.

Table 5: License of each dataset we used for acoustic unit investigation, pretraining, and fine-tuning.

| Dataset | Unit Analysis | Pretraining | Fine-tuning | License |
|---|---|---|---|---|
| CREMA-D [65] | ✓ | ✗ | ✗ | Open Database License |
| Multilingual LibriSpeech [71] | ✓ | ✓ | ✗ | CC-BY-4.0 |
| People's Speech [75] | ✗ | ✓ | ✗ | CC-BY-SA |
| GigaSpeech [72] | ✓ | ✓ | ✗ | Apache-2.0 |
| Common Voice 15.0 [76] | ✗ | ✓ | ✗ | CC-0 |
| Voxpopuli [77] | ✗ | ✓ | ✗ | CC-0 |
| DailyTalk [70] | ✗ | ✗ | ✓ | CC-BY-SA 4.0 |
| Expresso [87] | ✗ | ✗ | ✓ | CC BY-NC 4.0 |
| Fisher [88] | ✗ | ✗ | ✓ | LDC User Agreement |

Table 6: Links to the model implementations, checkpoints, and libraries used.

| | Link |
|---|---|
| XLS-R-based Unit Extractor [64] | https://github.com/facebookresearch/seamless_communication |
| Mistral-7B [67] | https://huggingface.co/mistralai/Mistral-7B-v0.1 |
| SpeechGPT [25] | https://github.com/0nutation/SpeechGPT/tree/main/speechgpt |
| *whisper-large-v3* [78] | https://huggingface.co/openai/whisper-large-v3 |
| Multipack sampler for data packing | https://github.com/imoneoi/multipack_sampler |
| BigVGAN [74] | https://github.com/NVIDIA/BigVGAN |
| Metric Calculation (WER, METEOR, ROUGE-L) | https://github.com/huggingface/evaluate |

**Unit-Voicebox for USDM** This Unit-Voicebox is a speech decoder used to restore speech from the output unit sequence of our model and From Scratch. It is trained in the same manner as the aforementioned variant but utilizes reference speech during training and inference to enable zero-shot speech reconstruction. To consistently respond in the same voice in multi-turn dialog scenarios, we use the adaptive TTS model, Voicebox as our speech reconstruction model, and leverage the spoken response of the preceding turn as the reference speech.

**Voicebox for Cascaded** This is the TTS model for one of our baseline models, the Cascaded model. We set the TTS model of the Cascaded model to Voicebox, training it with the same data and method as unit-Voicebox for a fair comparison. Unlike the unit-Voicebox, which does not require a duration predictor, we train a separate feed-forward duration predictor following Le et al. [66].

## A.4  Additional Details for Evaluation

### A.4.1  Models, Datasets, Training Details

We list the models we used for each component of our model and the baselines in Table 4. In Table 4, 'LLM' refers to the language model used prior to performing speech-text pretraining and/or fine-tuning. Additionally, we list the licenses of the datasets used in Table 5, and include links to the open-source implementations, checkpoints, and packages we use in Table 6.

We utilize the DailyTalk dataset to evaluate the performance of USDM. We follow the train/test split of Lee et al. [70] and preprocess the data for single-turn spoken dialog. As a result, we obtain a total of 20,117 training samples and 1,058 test samples.

```
Below is a dialog between Speaker 1 and Speaker 2, and two possible responses from Speaker 2.
Choose more contextually appropriate response between them.
{previous dialog}
### Output A: {response_1}
### Output B: {response_2}
Which one is better, Output A or Output B?
[Additional Consideration]
    – Give a penalty to unnecessary repetitions.
    – There would be cases that reach maximum sequence length. Do not deduct score for these cases.
    – Give a penalty for unengaging and simplistic response.
    – Only write a single char as your answer, 'A' for Output A or 'B' for Output B.
    – Do not add any explanation.
Decision:
```

Figure 6: Prompt used for LLM-based evaluation utilizing GPT-4.

### A.4.2 Human Evaluation and GPT-4 Judge

As mentioned in Section 4.1.2, we conduct various human evaluations and GPT-4-based assessments [1]. First, we employ Amazon Mechanical Turk to perform human preference tests and evaluate prosody and naturalness through P-MOS and MOS. In the human preference test, as detailed in the main paper, we present evaluators with previous spoken dialog and spoken input along with two candidate spoken responses. Evaluators are asked to choose which response is more appropriate, considering comprehensive aspects such as content, prosody, and sound quality. We provided evaluators with the instruction, *"Given the spoken dialog of two speakers, which response is more suitable? Please consider comprehensive aspects such as content, speech quality, and prosody."* All comparative experiments are evaluated by 150 evaluators, respectively, and the total cost for these evaluations is approximately $200.

We also conduct a qualitative evaluation by measuring the 5-scale mean opinion score (MOS) and prosody mean opinion score (P-MOS), both ranging from 1 to 5 points. For the P-MOS, we provide evaluators with the input spoken dialog and the corresponding ground truth text response. They are then asked to listen to the speech matching the ground truth text response and evaluate the prosody, considering both the spoken dialog and the response text. Additionally, we measure the MOS to judge audio quality and naturalness. In this scenario, evaluators are given only the response text and its corresponding spoken response without any preceding spoken dialog and are asked to rate the audio quality and naturalness. The instructions provided for the P-MOS and MOS tests are: *"How natural is the prosody in this recording? Please focus on the prosody in the context of the spoken conversation flow and the given text response, and ignore other aspects such as speaker ID and sound quality."* and *"How natural (i.e., human-sounding) is this recording? Please focus on the audio quality and the naturalness of pronunciation."*, respectively. 198 evaluators participate in the P-MOS measurement, and another 176 participate in the MOS measurement. For all 5-scale evaluations, we provide examples of speech rated as 1, 3, and 5 points as a reference to guide evaluators. We spend a total of approximately $250 on these evaluations.

To evaluate the semantic quality of the audio generated by each model, we utilize transcripts obtained by passing the generated audio through a separate ASR model, *whisper-large-v3*. These transcripts are then evaluated using GPT-4 [1]. As introduced by Zheng et al. [81], we provide the GPT-4 model with evaluation instructions, previous dialog, and two candidate responses each from USDM and a baseline model for comparison, asking it to choose the preferred response. The template used for this evaluation is shown in Figure 6.

Among all available APIs, we use *gpt-4-0125-preview* for evaluation. To avoid bias due to the order of response candidates, we assess the responses from the two models in both their original and reversed orders. Preference is primarily judged based on content appropriateness, but penalties are assigned for unengaging responses such as simple short answers and fillers. If the results differ between the two evaluations, it is marked as a Tie, if both prefer Output A, then A, and if both prefer Output B, then B, with results in Table 2.

All $p$-values in the main text are obtained through the Wilcoxon signed-rank test. For the preference test, a score of 1 is assigned if a model is preferred, 0 for a tie, and -1 if not chosen. We then conduct a test using these scores and report the respective $p$-values.

Table 7: METEOR and ROUGE-L results measured using the text obtained from ASR of the spoken response (Transcribed Response) and results measured using the intermediate text response (Intermediate Response).

| Method | Transcribed Response | | | Intermediate Response | |
|---|---|---|---|---|---|
| | METEOR | ROUGE-L | TTS WER | METEOR | ROUGE-L |
| Ground Truth | – | – | 2.2% | – | – |
| USDM | 13.1 | 15.7 | 2.0% | 13.8 | 16.5 |
| Cascaded | 12.5 | 15.0 | 1.3% | 12.9 | 15.5 |
| From Scratch | 8.6 | 10.6 | 64.0% | 10.6 | 13.0 |
| SpeechGPT | 9.9 | 11.8 | 23.2% | 12.1 | 13.8 |

Table 8: Six types of speech-text interleaved sequences used to evaluate the performance of the pretrained model, along with the templates used for measuring PPL. For sequences with a continuation relationship, the speech and text data are split in half, combining one modality from the first half (e.g., `speech1 token` or `text1 token`) with the remaining modality from the second half (e.g., `text2 token` or `speech2 token`).

| Sequence | Template |
|---|---|
| Unconditional Text | `text token` |
| Unconditional Unit | `speech token` |
| Correspondence - Unit-to-Text | `speech token <|correspond|> text token` |
| Correspondence - Text-to-Unit | `text token <|correspond|> speech token` |
| Continuation - Unit-to-Text | `speech1 token <|continue|> text2 token` |
| Continuation - Text-to-Unit | `text1 token <|continue|> speech2 token` |

## A.5    Additional Experiments and Results

### A.5.1    Additional Results for DailyTalk

In Section 4.1.2, we assess the semantic quality by using the transcribed text of the spoken response generated by each model, using the *whisper-large-v3*, and measure METEOR and ROUGE-L scores. Our models and baselines first generate a text response either within the end-to-end pipeline or through a separate model. We also measure METEOR and ROUGE-L scores for these intermediate text responses on the test sets of DailyTalk, and the results are presented in Table 7.

Errors occur during the generation of the spoken response using intermediate text and the transcription of that spoken response for evaluation, leading to a performance gap between the results measured using the intermediate text response and the transcript of the spoken response. Despite the error gap, we confirm that USDM outperforms baselines in terms of the semantics of the intermediate response. Notably, a higher TTS WER increases the gap between the results based on the intermediate text response and the semantic performance of the final spoken response.

### A.5.2    Ablation Studies for Pretraining Scheme

In this section, we provide further explanation of the ablation studies on the pretraining schemes discussed in Section 4.2.1. We design interleaved sequences excluding each key relationship, continuation and correspondence, to demonstrate the necessity of each relationship within our proposed speech-text pretraining scheme. Additionally, we follow the cross-modal pretraining scheme proposed in Spectron [57], which we name Setup 3.

**Setup 1** We create interleaved speech-text sequences composed solely of continuation relationships and use these for speech-text pretraining. The interleaved sequences used for Setup 1 can be obtained by skipping the last step of the 3-step data preparation process described in Section 3.2. This approach is similar to previous works such as BLSP [44] and SpiRit-LM [56].

Table 9: PPL of various pretraining schemes for diverse unit and text combinations for the `test-clean` subset of LibriSpeech. T2U represents text-to-unit, and U2T represents unit-to-text, with PPL measured only for the subsequent modality. Lower is better.

| Method | Overall | | Unconditional | | Correspondence | | Continuation | |
|---|---|---|---|---|---|---|---|---|
| | Text | Unit | Text | Unit | U2T | T2U | U2T | T2U |
| Ours | 6.886 | 4.813 | 17.175 | 5.037 | 1.133 | 4.113 | 16.781 | 5.380 |
| Setup 1 | 14.485 | 5.261 | 17.195 | 5.047 | 11.578 | 5.345 | 15.267 | 5.398 |
| Setup 2 | 31.679 | 5.619 | 17.846 | 5.107 | 1.108 | 4.098 | 1607.743 | 6.600 |
| Setup 3 | 21.392 | 5.146 | 17.463 | 5.086 | 1.107 | 4.110 | 506.374 | 6.521 |

**Setup 2** We construct cross-modal sequences exclusively with correspondence relationships. The interleaved sequences with this relationship are typically formatted as ''`speech token <|correspond|> text token`'' and ''`text token <|correspond|> speech token`'' sequences, similar to SpeechGPT [25].

**Setup 3** We also compare a scheme that utilizes one fixed template for pretraining. Following Spectron [57] we pretrain using an interleaved sequence where the input speech is transcribed into text, followed by predicting the subsequent response text and synthesizing the corresponding speech. Assuming the speech and the corresponding text are split into two parts (speech1, text1, speech2, text2), we perform cross-modal pretraining using the interleaved sequence ''`speech1 token <|correspond|> text token <|correspond|> speech2 token`'', where the `text token` is obtained by concatenating text1 and text2.

Each model is trained with the same data and hyperparameters as our pretraining. All models have the same vocabulary size. We measure the PPL of various combinations of speech-text sequences created using the LibriSpeech `test-clean` subset. We create 6 types of interleaved sequences and the templates of these sequences are listed in Table 8. For measuring the PPL of text tokens, we normalize the probability by excluding the probability of units and use this normalized probability to measure the PPL. Similarly, for speech modality PPL, we compute the logits and probabilities for only the unit tokens, which have a vocabulary size of 10,000, and then calculate the PPL. To evaluate pretraining schemes where only one of the special tokens `<|correspond|>` or `<|continue|>` is used, we insert the special token used during pretraining at the boundary between the two modalities, regardless of the combination being evaluated.

The PPL for each combination is listed in Table 8. Setup 1 and Setup 2, which model only one relationship, fail to model the other and exhibit high PPL values. Additionally, Setup 3, which uses a specific fixed template of interleaved sequences for pretraining, shows superior performance in interleaved sequences with a correspondence relationship but is unable to model a continuation relationship. In contrast, our speech-text model, which universally models various relationships, demonstrates consistently powerful performance regardless of the sequence type.

### A.5.3 Per-Task Training Dynamic Analysis

Our model adopts an end-to-end pipeline with intermediate text, where the input speech is first transcribed, followed by generating the response text. Consequently, the model simultaneously learns unit-to-text, text response generation, and text-to-unit, with each task potentially reaching its optimal point at different epochs. To observe the training dynamics of each task, we train the USDM on the DailyTalk dataset for 5 epochs and monitor TTS WER, STT WER, METEOR, and ROUGE-L at each epoch. As shown on the left side of Figure 7, while STT WER remains consistent over each epoch, TTS WER, METEOR, and ROUGE-L scores improve, suggesting that the dialog modeling task and text-to-unit tasks are more challenging compared to the unit-to-text task.

Additionally, we build two spoken dialog models using Low-Rank Adaptor (LoRA) [90] for fine-tuning the pretrained speech-text model on the DailyTalk dataset, and observe similar tendencies. We fine-tune the model using a higher learning rate of $10^{-4}$, comparable to USDM, with LoRA ranks of 8 and 256. Consistently, as the right side of Figure 7 illustrates, increasing the number of fine-tuning parameters improves semantic performance in dialog and unit synthesis performance but deteriorates the performance in the unit-to-text task. Considering these observations and the analysis

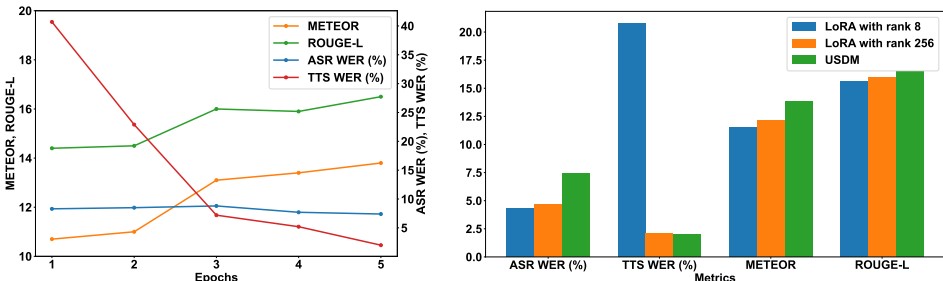

Figure 7: Left is the quantitative results for each epoch of the USDM fine-tuned on DailyTalk. The figure on the right illustrates the performance of the Spoken Dialog Model when trained with Low-Rank Adaptation (LoRA) versus full fine-tuning.

in Section 4.3 that demonstrates the importance of unit-to-text performance in USDM, we plan to explore strategies to mitigate overfitting in the unit-to-text task by varying the loss weight for each task within the pipeline or by applying the curriculum learning approach in future work.

### A.5.4 Additional Attention Maps

We plot the attention maps of the generated text response to the speech input and its model-generated transcript for 6 selected layers in Figure 4. Given that our spoken dialog model consists of 32 layers, we additionally include attention maps of another sample for all layers in Figure 8.

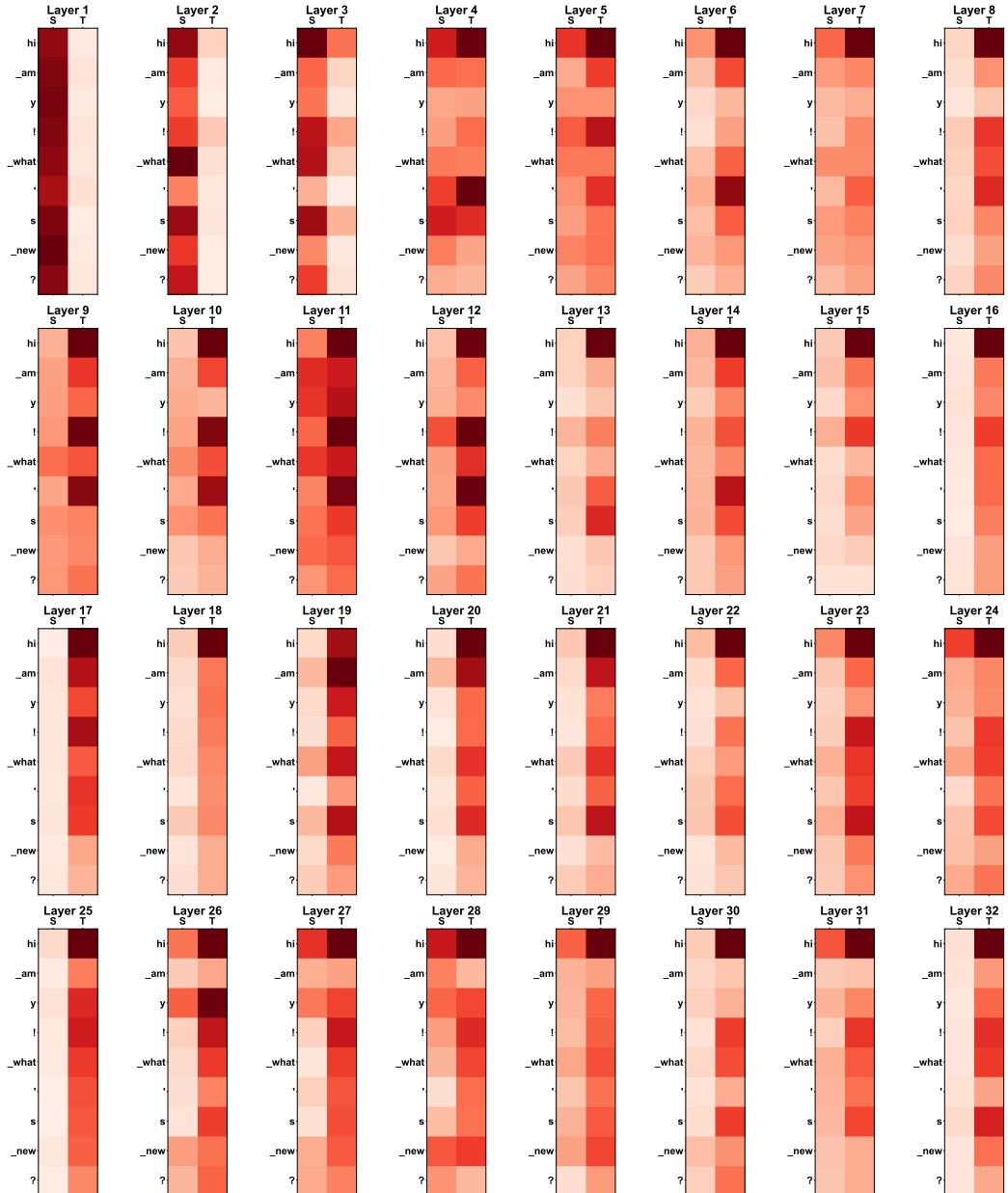

Figure 8: Attention map plots for the USDM response to the input speech "Hi George! It's good to see you!". We plot attention maps for all layers as described in Section 4.3. Although there are variations in intensity, we observe in all layers that the response text attends to both the speech input and the transcribed text.

