# OpenReview forum: "Paralinguistics-Aware Speech-Empowered Large Language Models for Natural Conversation"
_NeurIPS.cc/2024/Conference — NeurIPS 2024 poster_

### Official Review · Reviewer_yvFi · 2024-07-07

**Soundness:** 3
**Presentation:** 3
**Contribution:** 3
**Rating:** 6
**Confidence:** 5

**Summary:**

The authors present a new large language model (LLM) framework called the Unified Spoken Dialog Model (USDM) that can directly understand and generate spoken dialog responses with natural prosody. This is achieved by incorporating prosodic information into the speech tokens and using a multi-step spoken dialog template for fine-tuning. Both automatic and human evaluations on the DailyTalk dataset show that USDM outperforms previous models in generating natural-sounding spoken responses.

**Strengths:**

The authors demonstrate the superior performance of USDM over existing models on the DailyTalk dataset and validate their training methods through thorough analysis.

Key contributions include: a unified pretraining strategy for effectively modeling the relationship between speech and text, an extensive spoken dialog modeling framework using prosody-infusing encoders and decoders, and an LLM-based modeling strategy for generating natural and coherent dialog responses.

Their work establishes a foundation for speech-enabled chat-based LLMs, showcasing a prototype that enhances LLMs with speech interaction capabilities.

**Weaknesses:**

Comparative Analysis: Extend the comparative analysis to include a wider range of previous methods in addition to SpeechGPT. While SpeechGPT is a strong benchmark, a more comprehensive comparison with other relevant approaches would provide a clearer understanding of the USDM's relative strengths and weaknesses.

Dataset Diversity: Expand the evaluation to include multiple datasets beyond DailyTalk. Evaluating the model on a diverse range of datasets would offer a more robust assessment of its generalization capabilities and performance across different scenarios.

Emotional Control: Explore and discuss the potential for controlling the emotional expression of the generated responses in the proposed USDM model. Addressing this aspect would provide insights into the model's ability to adapt to different emotional contexts and potentially open avenues for future research in this direction.

**Questions:**

--

**Limitations:**

--

---

> ### Author Rebuttal · Authors · 2024-08-06
>
> First of all, thank you for your thoughtful comments, feedback, and questions. We provide explanations and answers to several questions below.
>
> **[W1] Comparative Analysis: Extend the comparative analysis to include a wider range of previous methods in addition to SpeechGPT.**
>
> Thank you for your valuable suggestions. Following your feedback, we have added an additional open source baseline for comparison on the DailyTalk dataset. The baseline we added is AnyGPT [1], a multimodal LLM capable of handling text, speech, images, and music as both input and output. We fine-tuned this model using its official implementations and pretrained checkpoints with a 7B parameter size on the DailyTalk dataset.
>
> The results of comparing this model with our model are presented in the table below. The comparison was conducted using the same metrics and test set as for Table 1 in the main paper. As shown in the table below, our model demonstrates superior performance compared to AnyGPT.
>
> **<Overall>**
>
> | **Method**     | **win**   | **tie**  | **lose**  | **MOS**         | **P-MOS**       |
> |----------------|-----------|----------|-----------|-----------------|-----------------|
> | Ground Truth | 45.9% | 8.0% | 46.1% | 4.45±0.05  | 4.41±0.05  |
> | USDM       | -     | -    | -     | 4.32±0.06  | 4.28±0.07  |
> | AnyGPT [1] | 65.6% | 5.1% | 29.3% | 3.54±0.07  | 3.49±0.08  |
>
> **<Semantic>**
>
> | **Method**     | **win**   | **tie**  | **lose**  | **METEOR**      | **ROUGE-L**     |
> |----------------|-----------|----------|-----------|-----------------|-----------------|
> | Ground Truth | 32.7% | 19.6% | 47.7% | -          | -           |
> | USDM       | -     | -    | -     | 13.1        | 15.7        |
> | AnyGPT [1] | 58.9% | 15.6%| 25.5% | 9.8         | 11.7        |
>
> Although AnyGPT used approximately 60,000 hours of speech and text data for pretraining, it also incorporated two additional modalities (images and music) into a single model. This, combined with its simpler speech-text pretraining scheme and less consideration of paralinguistic information in a multimodal LLM, contributes to its inferior performance compared to USDM.
>
> **[W2] Dataset Diversity: Expand the evaluation to include multiple datasets beyond DailyTalk.**
>
> Thank you for your valuable feedback. When preparing our model, we considered various datasets besides DailyTalk for evaluation. However, publicly available spoken dialog datasets were very limited. Among the candidates, the Fisher dataset, known for its frequent fillers, backchannels, and simple unengaging responses, posed significant challenges for model evaluation from a semantic perspective. Additionally, this dataset, collected in the early 2000s, comprises 8kHz low-quality telephone conversations, making it challenging to discern paralinguistics. Hence, we decided not to include it in our comparative analysis in the paper and used it only for research demonstration purposes on our demo page.
>
> Since the submission of our paper, a new spoken dialog dataset called MultiDialog [2] has released. Along with the valuable suggestions from other reviewers, our next step will be to validate our model's capability across more diverse languages, various tasks, and a broader range of datasets, including MultiDialog. Thank you once again for your insightful suggestions.
>
> **[W3] Emotional Control: Explore and discuss the potential for controlling the emotional expression of the generated responses in the proposed USDM model.**
>
> Thank you for your valuable suggestions. Following your advice, we have made slight modifications to our model's training template in Figure 5 of the main paper to enable emotion control. Originally, our template for spoken dialog was structured as speech1 $\rightarrow$ text1 $\rightarrow$ text2 $\rightarrow$ speech2. To explicitly control emotion, we modified it to speech1 $\rightarrow$ text1 <emotion1> $\rightarrow$ <emotion2> text2 $\rightarrow$ speech2.
>
> By using this modified template to train USDM, we can set <emotion2> to our desired emotion during inference, allowing us to control emotional expression. We preprocessed the MultiDialog dataset using this template and trained USDM accordingly. We have updated our demo page with speech samples generated by this model, demonstrating the possibility of USDM controlling emotional expression.
>
> While we demonstrated the potential, we observed several limitations in the MultiDialog dataset that hinder effective emotion control for spoken responses. MultiDialog is designed with scripts and corresponding ground truth emotions provided to participants, who then act out the dialogs. This dataset includes text, audio, and video (talking face) components. Because the recordings were made by acting out the provided emotions and scripts, we observed that the MultiDialog dataset often lacks precise alignment between speech and ground truth emotion labels. In some instances, the emotion is reflected in the video or transcript but not in the speech. This misalignment results in generated samples where emotions are only reflected in the text response or not prominently infused. Additionally, beyond dataset issues, we observed cases where the model struggled to control emotion when the desired emotion did not match the conversational context.
>
> We believe that as more high-quality datasets with well-aligned emotional expressions become available, the performance of our model will improve. Furthermore, we plan to propose additional techniques to disentangle the context of the conversation from the emotion, thereby enhancing emotional controllability. We appreciate your valuable feedback and look forward to incorporating it into our future work.
>
> **[Reference]**
>
> [1] Anygpt: Unified multimodal llm with discrete sequence modeling. (2024).
>
> [2] Let's Go Real Talk: Spoken Dialogue Model for Face-to-Face Conversation. (2024).

---

> > ### Comment · Area_Chair_NSFX · 2024-08-13
> > **Official Comment by Reviewer yvFi (made visible to authors)**
> >
> > "Dear Authors,
> >
> > Thank you for addressing my comments and providing detailed responses to each of the points raised. I appreciate the effort you’ve put into refining your paper. Below are my thoughts on the updated content:
> >
> > **[W1] Comparative Analysis:**
> >
> > The addition of AnyGPT [1] to your comparative analysis is a valuable improvement.
> >
> > **[W2] Dataset Diversity:**
> >
> > Your explanation regarding the challenges of incorporating datasets like Fisher and the choice to focus on the DailyTalk dataset is understandable. The inclusion of the MultiDialog [2] dataset in future evaluations is a commendable step towards assessing your model’s generalizability and performance across diverse scenarios. I look forward to seeing the results from these additional datasets as they become available.
> >
> > **[W3] Emotional Control:**
> >
> > The modifications made to enable emotion control in the USDM model are promising.
> >
> > Overall, your responses have addressed the concerns raised comprehensively."

---

### Official Review · Reviewer_BwMS · 2024-07-08

**Soundness:** 3
**Presentation:** 4
**Contribution:** 3
**Rating:** 7
**Confidence:** 4

**Summary:**

This paper introduces a method for modeling spoken dialog which relies on a speech-text LLM, pretrained with a combination or text and discrete speech tokens which capture semantics as well as prosody. The pretraining regime attempts to get the LLM to capture two types of relations between text and speech tokens: continuation and correspondence.
The approach is evaluated on the DailyTalk dataset using both automatic metrics and human preference judgements, and is shown to outperform a cascaded approach relying on ASR and TTS, as well as other baselines.

**Strengths:**

- The pretraining approach is simple but ingenious in terms of modeling.
- Speech units are pre-evaluated for paralinguistic content by testing them on emotion classification, and on speech reconstruction.
- Aspects of the pretraining scheme are evaluated via ablations.

**Weaknesses:**

The main weakness that the method relies on a massive amount (~10 years) of transcribed English speech. This makes it limited in its applicability to just a handful of very resource rich languages.

The evaluation relies on a single dataset in a single language.

The title places the focus of the work on integrating paralinguistic information, but in the actual paper this is only one, and not the most salient, part of the framework. The prosody is aspect is evaluated via the P-MOS score, but not analyzed in depth.

**Questions:**

Do you have any insight into which specific aspects of prosody this approach captures, as compared to the baselines?

**Limitations:**

Several limitations are clearly articulated. The reliance on large amounts of transcribed speech for training is not, however.

---

> ### Author Rebuttal · Authors · 2024-08-06
>
> Thank you for the constructive feedback. We provide our point-to-point response below.
>
> **[W1,2, L1] The main weakness that the method relies on a massive amount (~10 years) of transcribed English speech. This makes it limited in its applicability to just a handful of very resource rich languages, / The evaluation relies on a single dataset in a single language.**
>
> We share your concerns and acknowledge the limitations you pointed out. Like many other multimodal and audio (speech) LLMs, our model also utilizes over 10k hours of audio for training. This can indeed be a barrier when expanding to entirely new languages. As per your suggestion, we will include this in the Limitations section.
>
> Generally, apart from a few languages like English, most languages have only a few hundred hours of available speech data at most. Recent research in tasks such as speech translation and personalized speech synthesis shows that when extensive data is available for a specific language (typically English), it can significantly enhance the performance of low-resource languages by leveraging the dataset of data-rich languages [1, 2].
>
> Following your suggestion, we will extend our methodology to a multilingual setup in future research. By leveraging the data from resource-rich languages, we aim to boost the performance of low-resource languages and validate our approach across various languages and datasets. Thank you once again for pointing us in this meaningful direction.
>
> **[W3, Q1] The prosody is aspect is evaluated via the P-MOS score, but not analyzed in depth. / Do you have any insight into which specific aspects of prosody this approach captures, as compared to the baselines?**
>
> We acknowledge the limitations in our ability to precisely determine which aspects of prosody are captured in conversations. We also faced challenges in evaluating these aspects individually, which led us to rely on P-MOS for overall prosody assessment.
>
> We have conducted further analysis on the units we employed to infuse paralinguistics to indirectly demonstrate which aspects of prosody our approach focuses on. To achieve this, we checked whether our units encapsulate various aspects of prosody. We trained classifiers to assess whether the acoustic units contain information related to other prosodic aspects not covered in Section 3.1, such as gender, pitch, tempo, and energy.
>
> We used the TextrolSpeech dataset [3] to train these classifiers, employing the same structure as the emotion classifier described in the main manuscript. For gender classification, we used binary classes (male/female), and for the other three aspects, we performed ternary classification following the dataset's predefined classes. Below is the table with the statistics for the test set of each class and the classification results:
>
> | **Class** | **Test set (Total: 199)**            | **Probability of Random Guess** | **Acoustic Unit Classifier Accuracy** |
> |-----------|--------------------------------------|---------------------------|----------------------------------------|
> | Gender    | Male: 104 / Female: 95               | 50.8%                     | 83.4%                                  |
> | Pitch     | High: 68 / Normal: 67 / Low: 64      | 34.2%                     | 70.9%                                  |
> | Tempo     | High: 127 / Normal: 44 / Low: 28     | 63.8%                     | 82.4%                                  |
> | Energy    | High: 76 / Normal: 64 / Low: 59      | 38.2%                     | 64.8%                                  |
>
> The results in the table above confirm that our adopted acoustic units contain information related to gender, speed, and pitch, in addition to emotion. Additionally, for energy, our speech encoder is based on XLS-R, which normalizes input speech using mean and variance. As a result, while the trend in energy (e.g., position of peak value, lowest value) may be preserved, the absolute energy levels are affected. This indicates that the energy trend alone must be used to classify the energy of speech, leading to comparatively lower classification accuracy compared to other aspects.
>
> Note that the goal of this experiment is not to build highly accurate classifiers but to check which aspects of prosody are embedded in the units. With more data and additional techniques, the classification accuracies could be improved. By confirming the presence of these aspects in the acoustic units, we hope to provide a clearer understanding of which aspects of prosody our approach captures.
>
> **[Reference]**
>
> [1] Audiopalm: A large language model that can speak and listen. (2023).
>
> [2] XTTS: a Massively Multilingual Zero-Shot Text-to-Speech Model. (2024).
>
> [3] Textrolspeech: A text style control speech corpus with codec language text-to-speech models. (2024).

---

> > ### Comment · Reviewer_BwMS · 2024-08-14
> >
> > I'd like to acknowledge the authors' response and thank them for providing the additional analysis. As to my assessment ultimately it remains unchanged as it was largely positive already.

---

### Official Review · Reviewer_4xzH · 2024-07-09

**Soundness:** 3
**Presentation:** 3
**Contribution:** 3
**Rating:** 6
**Confidence:** 5

**Summary:**

This paper introduces an extensive speech-text LLM framework, the Unified Spoken Dialog Model (USDM), designed to generate coherent spoken responses with naturally occurring prosodic features relevant to the given input speech without relying on explicit automatic speech recognition (ASR) or text-to-speech (TTS) systems.

**Strengths:**

1)  This paper present how to integrate paralinguistics in speech-empowered LLMs. This topic is important and beneficial for both multimodal and speech communities.

2) The interleaved pre-training schedule is reasonable and effective that provides solution to mitigate modality gap between speech and text.

3) The finding in 3.1 is also interesting for speech tokenization.

**Weaknesses:**

1) Unfair comparison. Given that this paper focuses on paralinguistics, based on the findings in section 3.1, it is natural that SpeechGPT's 1k acoustic tokens would perform worse than USDM.

2) With such a reasonable pre-training process, it is regrettable that this work has not managed to extend the speech-text associations learned by the model to different tasks (such as [1]). The paralinguistics is important for human speech understanding beyond spoken dialog.

3) Some listening examples of zero-shot TTS (especially emotional TTS) are recommended to demonstrate that USDM can better mimic the emotion and prosody information from reference speech.

[1] Nguyen, Tu Anh, et al. "Spirit-lm: Interleaved spoken and written language model." arXiv preprint arXiv:2402.05755 (2024).

**Questions:**

1) How is the instruction-following capacity of USDM? Since this issue is unrelated to paralinguistics, it's ok to skip this question.

2) Is it possible for USDM to perform speech-to-speech directly with more data or training schedule? Such modality changing results in low efficiency. If can, which kind of strategy is required can be based on USDM?

3) Do authors have open-source plan for pre-trained model? I would like to increase the rating since such an interleaved speech-text foundation model can benefit to both speech and multimodal communities.

**Limitations:**

See weakness

---

> ### Author Rebuttal · Authors · 2024-08-06
>
> We appreciate your insightful comments and constructive questions. We address your concerns below.
>
> **[Q3] Do authors have open-source plan for pre-trained model?**
>
> Sure. We plan to release our code and pretrained models. Our model consists of a speech-text pretrained model, fine-tuned spoken dialog model (USDM), and unit-Voicebox, which restores units to speech. All three models will be made publicly available. In addition, since unit-Voicebox can perform speaker adaptation, we are currently preparing methods to prevent misuse (e.g., a classifier to distinguish synthesized speech, etc.). We intend to release unit-Voicebox along with these preventive measures.
>
> **[W1] Given that this paper focuses on paralinguistics, based on the findings in section 3.1, it is natural that SpeechGPT's 1k acoustic tokens would perform worse than USDM.**
>
> Our goal is to develop a model capable of natural conversation that reflects paralinguistics. In the context of spoken dialog modeling, we emphasize not only the incorporation of paralinguistic information but also the effective capture of semantic content within conversations.
>
> In Section 3.1, we highlighted that units with 10k acoustic tokens encapsulate pitch trends and emotional information from a paralinguistic perspective. Furthermore, in Section 3.2, we introduced a novel pretraining scheme aimed at capturing cross-modal relationships, which we believe is particularly beneficial for enhancing semantic understanding. The effectiveness of this scheme is demonstrated in Sections 4.1 and 4.2.1.
>
> For a more detailed comparison with SpeechGPT, we constructed an ablation setup, Setup 2, which is trained with a pretraining scheme similar to SpeechGPT but uses the same dataset and acoustic tokens as our approach. As shown in Table 3 of the main paper, Setup 2 performs worse than our approach. This comparison illustrates the efficacy of our pretraining scheme in maintaining semantic coherence, even with identical acoustic tokens. We hope this response clarifies our efforts to ensure a more precise and fair comparative analysis of the proposed components.
>
> **[W3] Some listening examples of zero-shot TTS (especially emotional TTS) are recommended to demonstrate that USDM can better mimic the emotion and prosody information from reference speech.**
>
> Thank you for your valuable suggestion. Following your idea, we have updated our demo page with several examples related to zero-shot TTS with several emotional speech prompts.
>
> **[W2, Q1] With such a reasonable pre-training process, it is regrettable that this work has not managed to extend the speech-text associations learned by the model to different tasks. / How is the instruction-following capacity of USDM?**
>
> Although we proposed and implemented speech-text pretraining for performing spoken dialog modeling, as you mentioned, our speech-text pretraining model can also be utilized for other speech-text downstream tasks. In this study, however, we intended to focus on spoken dialog modeling, as mentioned in our limitations section.
>
> Currently, our model is fine-tuned with a focus on the spoken dialog modeling task, so its instruction-following capability is not well demonstrated. Based on other existing studies [1,2], we believe that the instruction-following ability can be incorporated into our model by learning various speech tasks with appropriate instructions for each task.
>
> Following your suggestion, we plan to apply our method to a variety of tasks beyond spoken dialog modeling in future direction and strive to enhance the instruction-following capacity of our model. Thank you for your valuable suggestions.
>
> **[Q2] Is it possible for USDM to perform speech-to-speech directly with more data or training schedule? If can, which kind of strategy is required can be based on USDM?**
>
> As you pointed out, USDM generates spoken responses through intermediate text to achieve better performance, which is shown in Table 3 of the main text (compared to S1 $\rightarrow$ S2). Recently, it has been demonstrated that the amount of data is crucial for performance, not only in speech [3] but also in various modalities (text, image, etc.) [4]. Therefore, as you mentioned, we believe that using more data will likely improve the performance of speech-to-speech direct modeling.
>
> To illustrate this point, we attempted direct spoken dialog modeling using only 10\% of the data used in the S1 $\rightarrow$ S2 experiment in Table 3 of the main paper. When comparing the results, the original METEOR score was 6.5 and ROUGE-L score was 7.7, while with reduced data, the METEOR score dropped to 5.1 and ROUGE-L score to 6.4. Although we have not tested the upper bound of performance with increased data due to the lack of large-scale high-quality spoken dialog datasets, we expect that increasing the data alone would improve the performance of direct spoken dialog modeling.
>
> Additionally, considering potential strategies, recent works have shown that training various tasks with a single model creates synergy, resulting in performance improvements compared to task-specific models [5, 6]. Therefore, by exploring and incorporating various tasks and techniques that can aid in direct speech-to-speech spoken response generation, we believe USDM will be able to perform spoken dialogues without the need for intermediate text.
>
> **[Reference]**
>
> [1] UniAudio 1.5: Large Language Model-driven Audio Codec is A Few-shot Audio Task Learner. (2024).
>
> [2] Audio flamingo: A novel audio language model with few-shot learning and dialogue abilities. (2024).
>
> [3] Audiobox: Unified audio generation with natural language prompts. (2023).
>
> [4] Scaling laws for neural language models. (2020).
>
> [5] u-LLaVA: Unifying Multi-Modal Tasks via Large Language Model. (2023).
>
> [6] UniverSLU: Universal Spoken Language Understanding for Diverse Tasks with Natural Language Instructions. (2024).

---

> ### Comment · Reviewer_4xzH · 2024-08-07
> **Improve My Rating**
>
> I appreciate the author's response, which has alleviated most of my concerns. Therefore, I decide to raise my rating to 6 as I mentioned.

---

### Author Rebuttal · Authors · 2024-08-06

We would like to extend our sincere gratitude to all the reviewers for their insightful comments. Before addressing each reviewer's concerns, we want to clarify a point that may have caused some confusion caused by our paper's title, "Integrating Paralinguistics in Speech-Empowered Large Language Models for Natural Conversation."

The primary goal of our research is to propose an approach for modeling spoken dialogs that accurately reflect both paralinguistics and content, thereby building a spoken dialog model. Therefore, our study focuses not only on paralinguistics but also on semantic coherence in spoken conversation. To achieve both, we analyzed acoustic tokens in terms of paralinguistics and adopted prosody-infused speech tokens in Section 3.1, and proposed an effective and novel speech-text pretraining scheme in Section 3.2. We hope this clarifies that our focus is on both paralinguistics and appropriate content in spoken dialog modeling. If our title continues to cause confusion, we are open to considering a change in the title.

---

### Decision · Program_Chairs · 2024-09-25

**Decision:**

Accept (poster)

**Comment:**

This work builds a speech-text LLM focused on speech-in/out generation of spoken dialogue turns. Due to its use of speech units with validated paralinguistic attributes, the model exhibits subjective prosodic appropriateness similar to the ground truth. Via a suite of cross-modal correspondence and continuation tasks and expanded training data, the model also performs well on response semantic appropriateness versus past works’ training schemes. I and the reviewers find the system a solid step towards speech-interactive LLMs. Given the nascence of the task, minor questions around eval fairness and granularity arose but were addressed with finer evaluations or already-existing ablations. **I recommend Acceptance**.

As both reviewers and authors acknowledge, the paralinguistic modeling is more incidental / coarsely evaluated (though the rebuttal to R2 improves this, as well as examples of emotional TTS/control if provided as R1, R3 request). **I suggest the authors change the title** to emphasize that paralinguistic coherence is more emergent than deliberate (e.g., “Integrating Paralinguistics in” —> “Paralinguistics-Aware”), and optionally to better reflect other aspects like the pre-training objective or the focus on multi-turn/holistic dialogue evals versus prior work.